# Androgen Receptor Binding Category Prediction with Deep Neural Networks and Structure-, Ligand-, and Statistically Based Features

**DOI:** 10.3390/molecules26051285

**Published:** 2021-02-26

**Authors:** Alfonso T. García-Sosa

**Affiliations:** Institute of Chemistry, University of Tartu, Ravila 14a, 54011 Tartu, Estonia; alfonsog@ut.ee; Tel.: +372-737-5270

**Keywords:** machine learning, artificial intelligence, androgen receptor, random forest, deep neural network

## Abstract

Substances that can modify the androgen receptor pathway in humans and animals are entering the environment and food chain with the proven ability to disrupt hormonal systems and leading to toxicity and adverse effects on reproduction, brain development, and prostate cancer, among others. State-of-the-art databases with experimental data of human, chimp, and rat effects by chemicals have been used to build machine-learning classifiers and regressors and to evaluate these on independent sets. Different featurizations, algorithms, and protein structures lead to different results, with deep neural networks (DNNs) on user-defined physicochemically relevant features developed for this work outperforming graph convolutional, random forest, and large featurizations. The results show that these user-provided structure-, ligand-, and statistically based features and specific DNNs provided the best results as determined by AUC (0.87), MCC (0.47), and other metrics and by their interpretability and chemical meaning of the descriptors/features. In addition, the same features in the DNN method performed better than in a multivariate logistic model: validation MCC = 0.468 and training MCC = 0.868 for the present work compared to evaluation set MCC = 0.2036 and training set MCC = 0.5364 for the multivariate logistic regression on the full, unbalanced set. Techniques of this type may improve AR and toxicity description and prediction, improving assessment and design of compounds. Source code and data are available on github.

## 1. Introduction

Concerns are rising over endocrine disruptors entering the environment and food chain [1,2,3,4]. The Androgen Receptor (AR) is a protein involved in reproduction, brain development, prostate cancer, androgen insensitivity syndromes, spinal and bulbar muscular atrophy, acne, and alopecia [5]. Androgen receptor pathway modulators are compounds that can have an effect on tumors and reproductive systems [1,2,3,4]. The CoMPARA challenge was a collaborative modeling effort to predict possible AR modulators based on a wide collection of state-of-the-art experimental data [6]. Different modeling techniques have been attempted, including molecular docking [7], support vector machines (SVMs), combined structure-based, ligand-based fingerprint distances to known compounds, and Naïve Bayesians [8], among others [6].

Toxicity modeling of compounds is important in several ways: compounds that are used in pharmaceutical and industrial applications need to be assessed for possible adverse effects on humans and other organisms, as well as being an important development barrier for new drugs and useful compounds [1,2,3,4]. Difficulties include the lack of experimental tests, including chronic and different exposure effects, as well as those of metabolites of compounds [1,2,3,4].

Machine learning (ML) and artificial intelligence (AI) are transforming many fields, including the computational chemistry and medicinal chemistry fields [9,10,11,12,13,14]. Particular advantages may be realized by ML methods in classification techniques for drug compound analysis and design [10,15]. A common criticism of ML methods is the potential to include a high number of variables that can have little insight into their physicochemical meaning [10]. Some ML and AI models are being developed with the aim of being “explainable” [16] and afford interpretability to chemical groups responsible for positive or negative contributions to a prediction.

The present work shows that different modeling techniques can have their advantages and disadvantages for modeling AR modulating compounds. Deep neural networks (DNNs) and graph convolutional neural networks (CNNs) have been used in other modeling studies, usually using featurization included in widely available packages [17]. Here, an effort was made to build Random Forest (RF) and DNNs with a given set of features that are chemically important based on calculated protein–ligand binding to several targets, chemical fingerprint distances, and other results from statistical techniques. The aim of the present work is to generate AR binding classification models by DNNs with different featurizations and compare them to other methods such as RF, CNN, multivariate logistic regression, regressors, and experimental data. Improving the predicted categorization of chemical compounds as AR binders using physicochemically and biologically relevant features can help in flagging molecules that may have the potential to disrupt AR pathways, and thus, may have the potential of toxic effects. In silico prediction of these effects is important given the reduction of animal testing, and the expense of testing, as well as a first, fast complement to testing.

## 2. Results

### 2.1. Balancing and Initial Separations

The provided training dataset in the CoMPARA challenge was highly unbalanced with a large number of nonbinders (N = 1 468) compared to binders (N = 205). Bias in datasets can affect strongly the results of ML algorithms and so addressing these issues is recommended [18]. Balancing of the training set was thus performed to provide an equal amount of binders and nonbinders.

Another effect was also apparent after docking calculations for the chimp protein (Figure 1), where distributions of docking scores for chimp androgen receptor for binders and nonbinders show that, for those compounds that have a nonzero docking score (N = 1310), the docking scores are stronger for binders than for nonbinders. However, for the smaller amount of those compounds with a docking score of zero (N = 363), the simple docking score on its own cannot separate both distributions.

A similar effect is seen for the calculated Bayesian probabilities (Figure 2), where the distribution for binders is higher in probability to belong to the normal distribution of known binders and vice versa for nonbinders. Bayesians constructed on the docking scores for both groups showed this separation with means and standard deviations of μ = −8.91 kcal/mol and σ = 1.94 for binders, and μ = −5.97 kcal/mol and σ = 2.01 for nonbinders.

The distribution of Tanimoto distances to average values for ECFP fingerprints of binders and nonbinders (Figure 3) shows skews in the distributions with nonbinders tending to have a tail skewed towards larger distances to the average of known binders, whilst the binding compounds have slightly larger distances to the average of known nonbinding compounds.

After balancing, the total numbers for both sets of compounds were: Number of compounds in train set: 410, composed of 205 binders and 205 nonbinders. No balancing was made for the evaluation set, where the number of compounds in validation set was 3882.

### 2.2. Comparison to RF and CNN

Classifier and regression ML models were built and evaluated (Table 1). For the Random Forest (RF) Classifier (I), the best results obtained after eightfold cross-validation were: Best hyperparameters: (100, ‘sqrt’), giving AUC values of train_score: 1.000000, validation_score: 0.7564. The Graph Convolutional Neural Network Classifier (CNN, VII) using hyperparameters: {‘learning_rate’: 0.0001403, ‘weight_decay_penalty’: 2.95 × 10^−6^, ‘nb_epoch’: 40}, gave AUC values of: train_score: 0.827864, validation_score: 0.739908. There is clearly overfitting occurring in RF and CNN methods given the high ROC-AUC values for the training set (Figure 4), with the best metrics for the evaluation score being around 19 runs.

### 2.3. DNN with Different Featurizations

The best result for the training and validation sets based on the ROC-AUC metric was the DNN using the supplied 12 features (Table 1, model II), and this model also showed less overfitting.

The models for regression did not achieve good *R^2^* values, which is logical due to the awkwardness of fitting a regression to a binary outcome value. For RF in regression mode: Best hyperparameters: (10, ‘log2’), giving *R^2^* values of train_score: 0.8817, validation_score: −0.0520. For CNN in regression mode, after fourfold cross-validation, with the best hyperparameters: {‘learning_rate’: 0.000359206871754689, ‘weight_decay_penalty’: 8.830664294504987 × 10^−6^, ‘nb_epoch’: 20}, *R^2^* values were: train_score: 0.2721: validation_score: −0.1926.

Increasing the number of estimators for RF results in increasing the degree of overfitting for the balanced training set (Figure 4). Tests on the full (unbalanced) initial training set show less overfitting, as well as using DNN (Figure 5).

DNN were better predictors than RF models for a variety of metrics (Figure 5). The results obtained are good for ROC-AUC, accuracy, F1-scores, and precision metrics for training (balanced), validation, as well as full training set (Table 1 and out.txt at https://github.com/AlfonsoTGarcia-Sosa/ML (accessed on 26 February 2021)). The Matthews correlation coefficients (MCC) scores obtained are reasonable, considering the lack of balance in the datasets, as well as the lack of distinction in features between binders and nonbinders in the evaluation set. This lack of feature distinction can be seen in Figure 6 and Figure 7, where t-maps, histograms, and density maps for calculated descriptors show a highly overlapping distribution for binders as well as nonbinders in the evaluation set, highlighting the difficulty of classification for this dataset (octanol/water partition (clog*P*), topological polar surface area (TPSA, Å^2^), number of heterocycles, molecular weight (MW, g/mol), number of rotational bonds (nRotB), number of hydrogen bond donors (nHBDon), number of hydrogen bond acceptors (nHBAcc), and Alog*P*).

### 2.4. Validation of Docking and Comparison to Other Methods

Decoys were generated for the androgen receptor using the DUD-E database (http://wiki.bkslab.org/index.php/DUDE (accessed on 14 February 2021)) and Receiver-Operator Curves (ROCs) and Area Under the Curves (AUCs) for the Human, Chimp, and Rat androgen receptor docking scores were calculated and plotted (Figure 8), showing a good separation of true positive from false positives in the most important, i.e., initial parts of the curves. Their values are high, and the chimp protein again shows that it is the most suited with an AUC of 0.832, and an enrichment factor at 1% of 68.92. AUC for human was 0.797, and AUC for rat was 0.744.

Comparing our results to a structure-based approach by Trisciuzzi et al. [7], they obtained the highest AUC of 0.76 for structures 2pnu and 2hvc, compared to 0.83 for the Chimp AUC in this work.

Using 20 gold-standard reference androgen receptor probe compounds as used by Kleinstreuer et al. [19] (Table 2) shows that there was a good result for predictions of 16/20, i.e., 80% were predicted correct for being a binder to AR (very weak binders were considered as nonbinders).

With respect to well-known compounds that are frequently misclassified [20], the results provided here (Table 3) show that four out of 11 compounds correctly predicted compared to three out of 11 reported elsewhere [20], the difference being the correct prediction of finasteride [20]. Chemical structures in Table 2 and Table 3 show several steroid core structures that may be difficult for algorithms to distinguish between actives and inactives, given the strong chemical similarity between them.

The loss function diagram for the best DNN (Figure 9, II) shows a relatively stable function with loss for the training set starting around 0.5 and fluctuating moderately up to 0.6 but then decreasing steadily to around 0.27, while the loss function for the validation set fluctuates moderately starting from 0.3 to 0.26.

The approach of using physicochemically and biochemically relevant user-defined features (12 features, II) is seen in the better metrics performed by the DNN trained and validated on these features rather than DNN trained on cddd descriptors (around 500 features, VI), as well as the closely performing RF on the user-defined features (I), and also being better than the CNN using vector featurization of the molecular graph (VII). In addition, the use of ML is warranted in this case, since the same features used in a multivariate logistic regression fashion produced metrics that were not as good, evaluation set MCC = 0.468 and training MCC = 0.868 for the present work compared to evaluation set MCC = 0.2036 and training set MCC = 0.5364 for the multivariate logistic regression on the full, unbalanced set [8].

## 3. Discussion

Toxicity classification problems can benefit from using DL and specific features that have rationalization on the biochemically and chemically relevant features of the compounds. In this case, the best results were provided by predicted binding category to chimp, rat, and human androgen receptor structures, in addition to average Tanimoto distances to known binders and nonbinders, as well as Naïve Bayesians as user-provided features to a DNN,. Unbalanced datasets can be transformed to balanced sets by unbiased case dropping and thus perform better in training and evaluation metrics. In particular, it is hard to produce toxicity data, especially chronic data, for chemicals with animals and humans, and this lack of data can translate into unbalanced datasets and difficulties for classification and regression techniques. Bias in datasets can be treated with different approaches, such as undersampling [21], as well as distribution following [18]. However, it is clear that the availability of more data is beneficial for methods and interpretations of androgen receptor pathway modulators and their toxicity potential. In addition, the most relevant biological assay, be that chimp, rat or human, or their use in consensus, may provide the best experimental setup for classification data.

Evidently, there are considerations to be taken about how to classify kinetic data [22,23] in many cases of biological interest, e.g., in antibody interactions, complex formation steady-state is not reached [22]; to distinguish between binding sites; and that analyzing interaction data from biosensor instruments is based on the simplified assumption that larger biomolecules interactions are homogeneous [23]. Also, that for the CoMPARA challenge [6], the organizers (EPA) used the thresholds determined in the CERAPP project and applied them to AR concentration-response values (AC_50_) from the literature, using the following scheme among several possible:
Strong: Activity concentration < 0.09 μMModerate: Activity concentration 0.09–0.18 μMWeak: Activity concentration 0.18–20 μMVery Weak: Activity concentration 20–800 μMInactive: Activity concentration > 800 μM

The use of ML in the form of DNN with relevant, user-specified features on a balanced set provides better results as compared to the same features in a multivariate logistic fashion, as well as purely structure- or ligand-based approaches, as seen by better AUC, MCC, and other values.

## 4. Materials and Methods

### 4.1. Training Set

The training set of compounds was provided during the CoMPARA challenge for predicting androgen receptor activity for chemicals [6], and included curated data with SMILES strings. This training set was composed of state-of-the-art experimental data from ToxCast [24], Tox21 [25], and DrugBank [26] databases, amounting to 1673 chemical compounds with 205 positives (binders), and 1468 negatives (nonbinders). Binders were coded as actives (“1”), nonbinders were coded as inactives (“0”). The SMILES strings were used as present in the files. Given that the training set was heavily unbalanced, the training set was balanced using pandas tools v. 0.25.3 [27].

### 4.2. Independent Evaluation Set

The evaluation data set was also provided in the CoMPARA challenge [6] from different databases being completely independent from the training set: EPA’s NCCT collected and curated PubChem data (64 sources) [6,28]. After including only binding data, there were 3882 compounds in the evaluation data set, composed of 446 positives (binders) and 3437 negatives (nonbinders). No balancing was performed for the evaluation set for an unbiased evaluation of the models.

### 4.3. Features

The structures for the human, chimp, and rat androgen receptor were downloaded from the PDB [29] (codes 3v49, 1t7r, and 3g0w) based on their resolution (1.4, 1.7, and 1.95 Å, resp.), completeness of sequence, and relevance of the complex. Protein X-ray crystal structures were preprocessed with the Protein Preparation Wizard from Schrödinger v. 2019 [30]. Docking scores were generated with Glide XP v. 2019 [31] centered on the orthosteric site of AR using 15 Å inner box and 40 Å outer search boxes, that differ from default settings. The results of each docking run were used as structure-based features: ‘HumDockScore’, ‘ChimpDockScore’, and ‘RatDockScore’ for the docking scores in kcal/mol of the human, chimp, and rat AR structures, respectively. The average of the docking scores for the three protein targets was also computed and stored as feature ‘AVG’ [32,33,34].

For ligand-based features, Extended connectivity fingerprints (ECFP), circular topology-based representations of compounds, were calculated with ChemAxon v. 2010 [35]. Distances between compound fingerprints were calculated by Tanimoto similarity using OpenBabel v. 2019 [36], giving ligand-based features named ‘avgD_Act’ and ‘avgD_Inact’, respectively, for the calculated distance to the average of known active and inactive compounds of the ECFP fingerprint for each compound.

Naïve Bayesians (NBs) were constructed using the means and standard deviations of the docking scores of actives to the chimp receptor, and the probability P given for each group ‘P_Act_dockChimp’, and ‘P_Inact’, respectively, were used as statistical features, as well as their ratio and Bayesian prediction (feature ‘PredBayes’) corresponding to which value of P calculated to each distribution was greater, with binding = 1, and nonbinding = 0, corresponding to ratio > 0.5 and ratio < 0.5, respectively.

Another feature created was ‘predMLogR’, the probability from a multivariate logistic classifier using these variables, as calculated in [8]. ‘PredBindingClass’ is also a feature, defined as the binary value for this predicted probability, i.e., ‘PredBindingClass’ = 1 if ‘predMLogR’ > 0.5, or else ‘PredBindingClass’ = 0. This is a distinct feature from the NB prediction above.

For comparison, the cddd group of latent-space encoded ligand-based descriptors was also used as described in the original publication [37].

### 4.4. Models

Three types of model were run: Deep Neural Networks (DNN), and for comparison, Random Forest (RF), and Graph Convolutional Neural Networks (CNN). Two types of featurization were used for DNN and RF: the cddd groups of ligand-based descriptors; and our own, user-specified features from structure-based (docking), ligand-based (fingerprint distances), and statistically based features. RF and DNN were run both as Classifiers (models I and II, respectively), and for comparison, as Regressors too (III and IV) with deepchem v. 2.3.0 [17]. Two types of featurizations were used: (1) user-specified features calculated for the compounds (‘HumDockScore’, ‘RatDockScore’, ‘ChimpDockScore’, ‘AVG’, ‘P_Act_dockChimp’, ‘P_Inact’, ‘PredBayes’, ‘ratio’, ‘avgD_Act’, ‘avgD_Inact’, ‘PredBindingClass’, ‘predMLogR’; see Section 4.3
*Features*, above); and (2) the cddd groups of ligand-based descriptors [37] (RF_cddd V, DNN_cddd, VI). The cddd featurization (512 features exploring the continuous descriptor space) have been reported to give good results for ML models for prediction of compound properties such as solubility and quantitative structure–activity relationships, as well as ligand-based virtual screening tasks [37]. CNN were also employed using atom-based featurization (VII). CNN models tend to be largely used for graphical data, with pixels or vector representations, for example. They have also been reported to give good results on compound property predictions; a study found that CNNs where atom properties are used instead of pixels are more accurate than DNN for predicting quantum chemical energies [10]. Batch size was 128 and 10-fold cross-validation was used, as well as the ROC-AUC as guiding metric. The models I and II probed features “max_features”: [“auto”, “sqrt”, “log2”, None]. Number of epochs was also varied for the DNN models, from 1 to 30,000,000.

### 4.5. Metrics

In all cases, the task classification or regressor was the “binding Class” status of the compounds, that is, coded 1 for binders and 0 for nonbinders that represented experimental actives and inactives for androgen receptor. Validation metrics included Area-Under-the Curve (AUC) measurements of the Receiver-Operator Curve of true positive and false positive rates, that range from 0 (complete misclassification) to 1.0 (complete classification), precision, recall, Matthews correlation coefficient (MCC), F1-score, and accuracy as determined by sklearn [38].

A confusion matrix has four fields: true positives (TP), true negatives (TN), false positives (FP), and false negatives (FN). Specificity (SP) is calculated as: TN/(TN+FP), Precision: TP/(TP+FP), Recall (sensitivity, SE): TP/(TP+FN), Accuracy (Acc.): (TP+TN)/(TP+FP+FN+TN), and MCC. MCC is calculated as:MCC = (TP∙TN − FP∙FN)/√((TN + FN)∙(TN + FP)∙(TP + FN)∙(TP + FP))(1)

F1 score is calculated as:F1 = 2 ((Precision × Recall))/((Precision + Recall))(2)

Dataset diversity analysis and visualization were performed with PUMA: Platform for Unified Molecular Analysis, Version 1.0 (Mexico City, Mexico, 2020) [39], as well as with t-map [40] that uses MHFP6 fingerprints [41].

All data and code were run on jupyter notebooks and python, deposited, and made available on github at: https://github.com/AlfonsoTGarcia-Sosa (accessed on 14 February 2021).

## 5. Conclusions

The aim of the present work was to generate androgen receptor (AR) binding classification models by deep neural networks (DNNs) with different featurizations and compare them to other methods such as random forests (RF), graph convolutional neural networks (CNN), multivariate logistic regression, regressors, and experimental data. DNNs with 12 user-specified structure-, ligand- and statistically based features were found to perform best at categorizing AR binders and nonbinders. They outperformed DNN with cddd features, as well as RF, and CNN methods, and regressors (expectedly, given the sharp category bins), as well as the same features in a multivariate logistical fashion, as well as simple docking. Implications are that explainability in machine learning (ML) features is important, as physicochemically and biologically relevant descriptors can perform best at the categorization for this particular AR dataset. In addition, different ML techniques may be best suited for different application tasks, with DNN performing better than RF given the overtraining seen in the RF models. CNN models may require more information, such as the protein-ligand binding pose or trajectories. The cddd featurizations may well perform better for property prediction and virtual screening. In the present work, the Chimp structure-based features performed better than other protein-derived features in this work and others published elsewhere. Improvement is still possible, given that the Matthews correlation coefficient (MCC) can be higher for the evaluation compounds even if the predictions obtained were good and improved on predictions for a golden standard of AR reference compounds. Better data, that is, less unbalanced and with better structural diversity, may help improve future predictions, as could be combining the present features with other ML and non-ML techniques, such as boosting or molecular dynamics simulations, respectively.

## Figures and Tables

**Figure 1 molecules-26-01285-f001:**
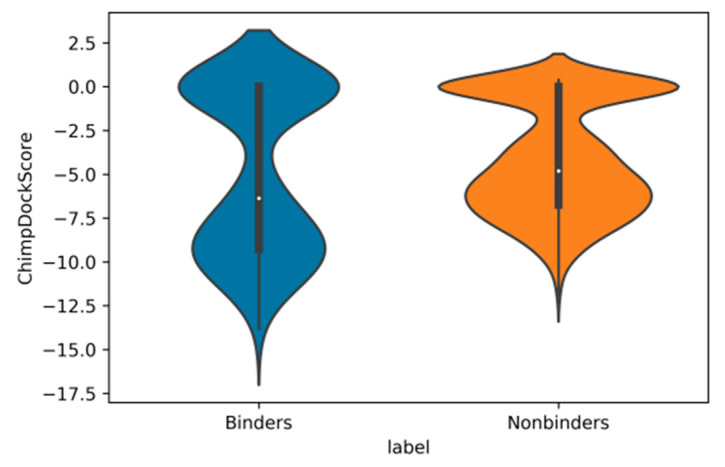
Violin plots of the distributions of docking scores (kcal/mol) with the chimp androgen receptor for binders (**blue**) and nonbinders (**orange**).

**Figure 2 molecules-26-01285-f002:**
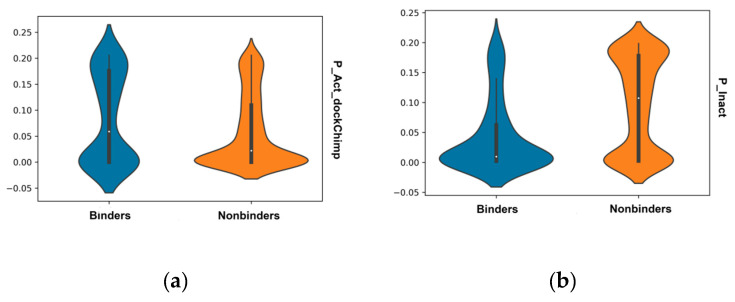
Violin plots for the distributions of Bayesian probabilities for docking scores for: (**a**) binders (P_act_dockChimp); and (**b**) nonbinders (P_Inact) for chimp androgen receptor docking scores for binders (blue) and nonbinders (orange).

**Figure 3 molecules-26-01285-f003:**
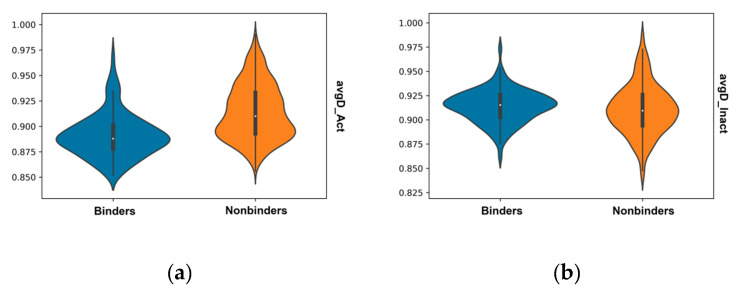
Violin plots of the distributions of average Tanimoto distances of binders (blue) and nonbinders (orange) to: (**a**) known binders (avgD_Act); and to: (**b**) known nonbinders (avgD_Inact).

**Figure 4 molecules-26-01285-f004:**
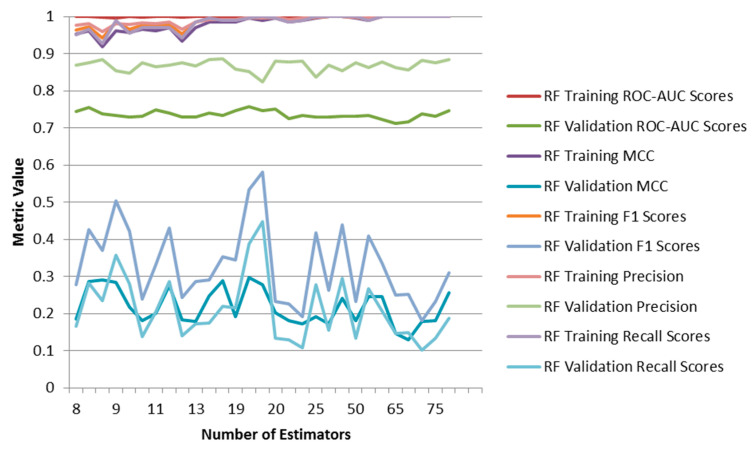
Metrics for different Random Forest models with varying number of epochs. MCC = Matthews correlation coefficients for balanced training test and validation test.

**Figure 5 molecules-26-01285-f005:**
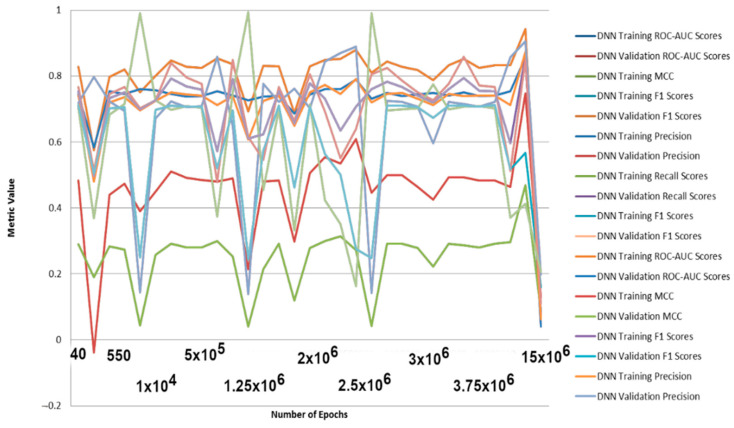
Metrics for different Deep Learning models with varying number of epochs. MCC = Matthews correlation coefficients for balanced training test and validation test.

**Figure 6 molecules-26-01285-f006:**
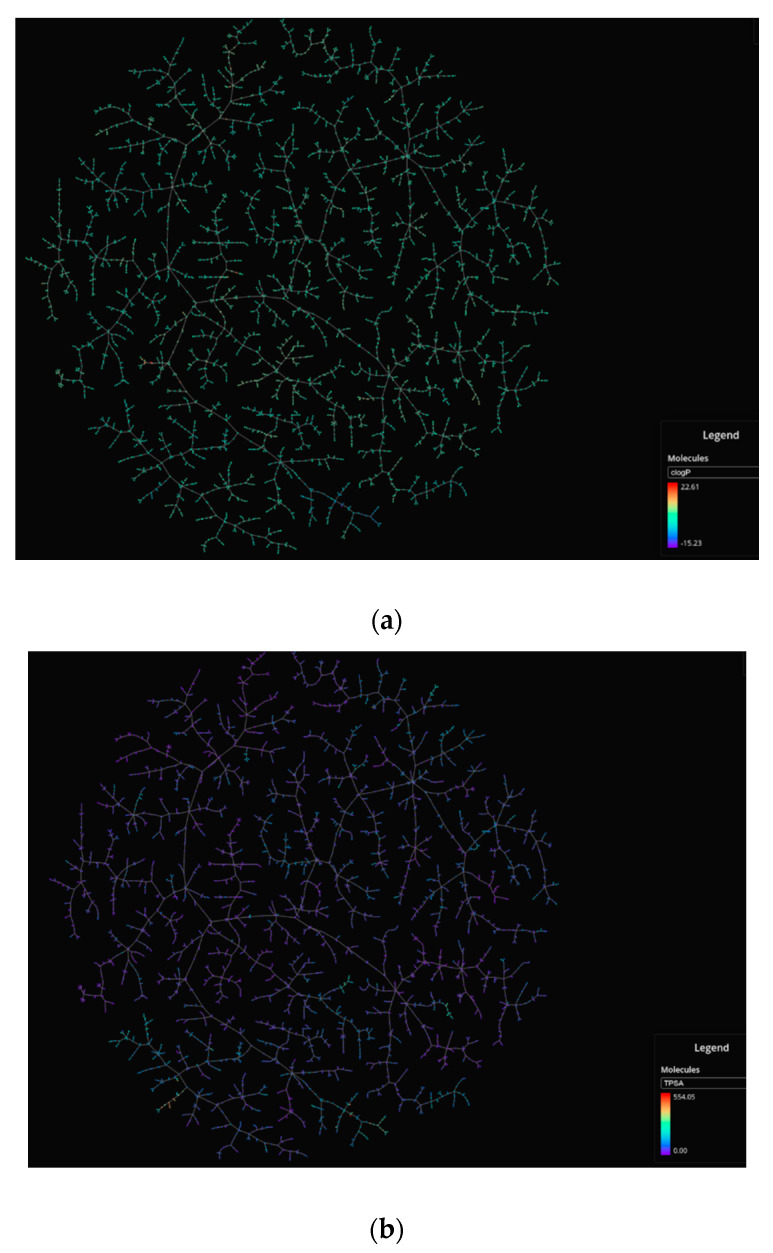
t-Maps for binders and nonbinders in the evaluation set. (**a**) clog*P*; (**b**) TPSA (Å^2^); (**c**) Number of heterocycles.

**Figure 7 molecules-26-01285-f007:**
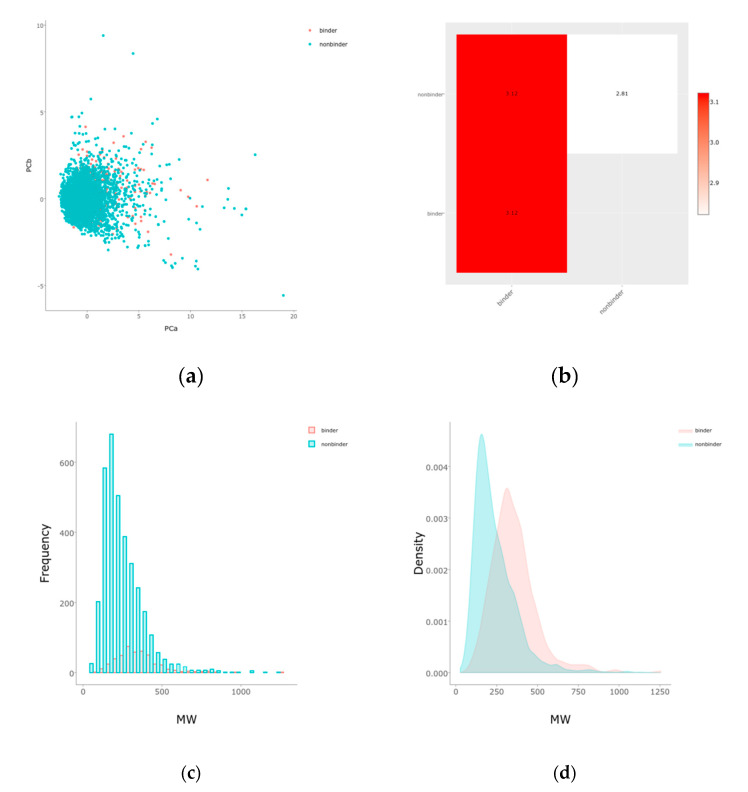
PUMA: (**a**) principal component analysis; (**b**) balance; and histograms and density plots for binders and nonbinders in the evaluation dataset according to: (**c**,**d**) MW (g/mol); (**e**,**f**) TPSA (Å^2^); (**g**,**h**) number of rotational bonds (nRotB); (**i**,**j**) number of hydrogen bond donors (nHBDon); (**k**,**l**) number of hydrogen bond acceptors (nHBAcc); and (**m**,**n**) Alog*P*.

**Figure 8 molecules-26-01285-f008:**
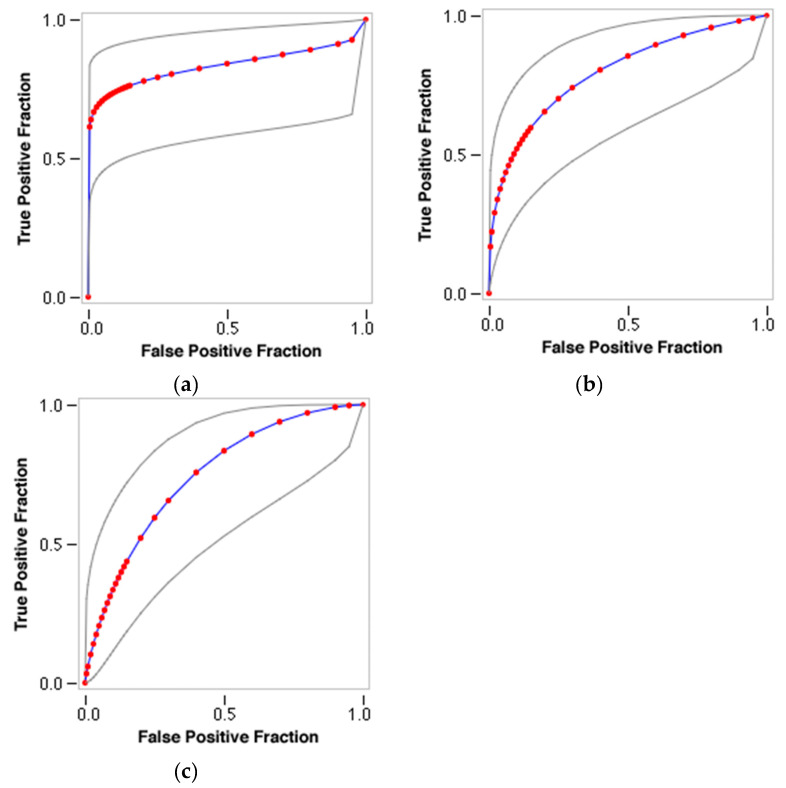
Receiver-Operator Curves (ROCs) and Area-Under-the Curves (AUCs) for the: (**a**) Chimp; (**b**) Human; and (**c**) Rat androgen receptor docking scores. Chimp AUC = 0.832; Human AUC = 0.797; and Rat AUC = 0.744.

**Figure 9 molecules-26-01285-f009:**
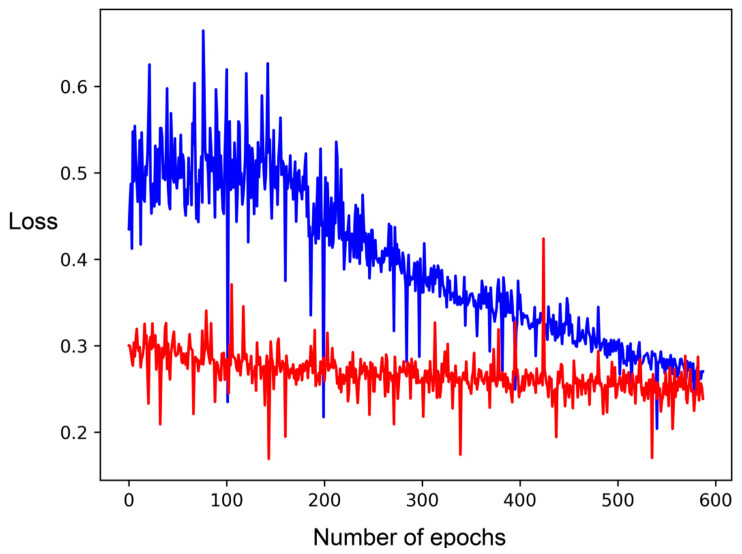
Loss functions of training (blue) and validation (red) datasets by number of epochs in DNN with user-defined features (II).

**Table 1 molecules-26-01285-t001:** Best methods obtained for different algorithms and featurizations. Standard deviations (s.d.) included for the best methods.

Method	Train ± s.d.	Valid ± s.d.	Best Hyperparameters
RF classifier myfeats (**I**)	AUC 0.9999 ± 0.0009; MCC 0.9951 ± 0.0153; F1 0.9963 ± 0.0153; Prec. 0.9976 ± 0.011; Recall 0.9951 ± 0.0198	AUC 0.7564 ± 0.0105; MCC 0.297435 ± 0.0478; F1 0.5805 ± 0.1041 (3 × 10^6^ epochs); Prec. 0.8856 ± 0.0148(1.5 × 10^5^ epochs); Recall 0.4481 ± 0.0866 (3 × 10^6^ epochs)	eightfold cross-validation, (19 runs, ‘sqrt’), 2.25 × 10^6^ epochs
DNN classifier myfeats (**II**)	AUC 0.9424 ± 0.0655; MCC 0.7472 ± 0.1283; F1 0.8608 ± 0.0754; Prec. 0.8732 ± 0.063; Recall 0.8585 ± 0.1092(4.5 × 10^6^ epochs)	AUC 0.8686 ± 0.0398; MCC 0.4685 ± 0.0892; F1 0.7943 ± 0.1617 (4.5 × 10^6^ epochs); Prec. 0.9052 ± 0.1988; Recall 0.8585 ± 0.2054 (4.5 × 10^6^ epochs)	Learning rate: 0.00047, weight decay penalty: 2.637 × 10^6^, 2.5 × 10^6^ epochs
GraphConv CNN (**VII**)	AUC 1.0	AUC 0.7264	−(50 runs, ‘sqrt’)
RF classifier CDDD features (**V**)	AUC 0.9997	AUC 0.7308	(18, ‘sqrt’)
DNN classifier CDDD features (**VI**)	AUC 0.8498	AUC 0.7563	Learning rate: 0.00067, weight decay penalty: 4.073 × 10^6^, 2.5 × 10^6^ epochs
RF regression myfeats (**III**)	*R*^2^ = 0.8817	*R*^2^ = −0.0520	(10 runs, ‘log2’)
DNN regression myfeats (**IV**)	*R*^2^ = 0.2721	*R*^2^ = −0.1926	fourfold cross-validation, learning rate: 0.000359 weight decay penalty: 8.831 × 10^6^, nb. epochs: 20

**Table 2 molecules-26-01285-t002:** AR pathway in vitro reference compounds and their predicted class according to DNN classifier (II).

CAS	Name	Structure	Agonist	Antagonist	Predicted	Correct
52806-53-8	hydroxyflutamide	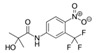	NA	Strong	1	Yes
90357-06-5	bicalutamide	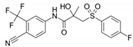	NA	Strong	1	Yes
122-14-5	fenitrothion	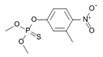	NA	Strong	0	X
63612-50-0	nilutamide	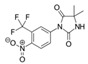	Negative	Moderate	0	X
427-51-0	cyproterone acetate	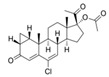	Weak	Moderate	1	Yes
80-05-7	bisphenol a	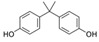	NA	Moderate/weak	1	Yes
330-55-2	linuron	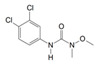	NA	Moderate/weak	0	X
13311-84-7	flutamide	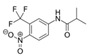	Negative	Moderate/weak	0	X
67747-09-5	prochloraz	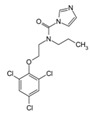	Negative	Moderate/weak	1	Yes
789-02-6	*o*,*p*′-ddt	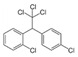	Negative	Weak	1	Yes
60168-88-9	fenarimol	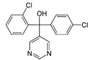	Negative	Very weak	0	Yes
58-18-4	methyl testosterone	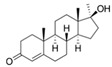	Strong	Negative	1	Yes
58-22-0	testosterone	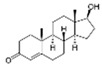	Strong	Negative	1	Yes
63-05-8	4-androstenedione	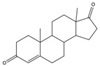	Moderate	Negative	1	Yes
1912-24-9	atrazine	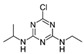	Negative	Negative	0	Yes
52918-63-5	deltamethrin	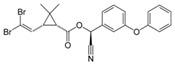	Negative	Negative	0	Yes
10161-33-8	17b-trenbolone	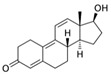	Strong	NA	1	Yes
797-63-7	levonorgestrel	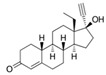	Strong	NA	1	Yes
68-22-4	norethindrone	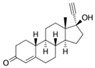	Strong	NA	1	Yes
521-18-6	5a-dihydrotestosterone	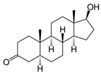	Strong	NA	1	Yes

**Table 3 molecules-26-01285-t003:** Chemicals That Were Frequently Predicted Inaccurately by Other Machine-Learning Models [20].

CAS	Name	Structure	Agonist	Antagonist	Predicted by II	Correct
58-18-4	methyl testosterone	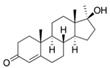	Strong	Negative	1	Yes
57-91-0	17α-estradiol	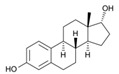	Inactive		1	X
63-05-8	4-androstenedione	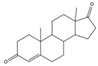	Moderate	Negative	1	Yes
486-66-8	daidzein	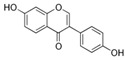		Inactive	1	X
98319-26-7	finasteride	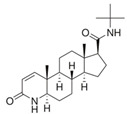		Inactive	0 (RF)	Yes
57-85-2	testosterone propionate	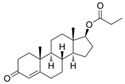	Strong	Inactive	1	X
51-28-5	2,4-dinitrophenol	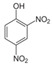	Negative	Negative	0	Yes
129453-61-8	fulvestrant	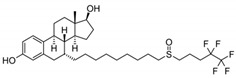		Inactive	1	X
84-74-2	dibutyl phtalate	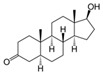	Active		0	X
117-81-7	diethylhexyl phtalate	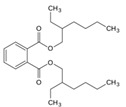		Active	0	X
13194-48-4	ethoprop	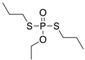		Active	0	X

## Data Availability

All of the data used is provided in text file format, all jupyter notebooks and python code used are also provided and are available at the public repository: https://github.com/AlfonsoTGarcia-Sosa (accessed on 14 February 2021).

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
