# Peer review of "Androgen Receptor Binding Category Prediction with Deep Neural Networks and Structure-, Ligand-, and Statistically Based Features"

_molecules, 2021, doi:10.3390/molecules26051285_

Round 1
Reviewer 1 Report
This is an interesting paper. Honestly, I am not a expert of ML and AI, as a androgen receptor researcher, I think the research is interesting and may have potential use in predicting androgen receptor agonist and antagonist. Unfortunately, think the paper can't be published as it is.
The author introduced and compared several prediction method, but no clear conclusion, even no results showed in abstract.
The results need some subtitles, at least a introduction sentence in each section is needed.
I strongly suggest the author add a conclusion paragraph at the end of the manuscript.
The discussion section needs rewrite, I understand the author did some of discussions in results section. I suggest the author rewrite the discussion and move the discussion sentence from results, or put the results and discussion together.
Author Response
Reviewer 1.1: This is an interesting paper. Honestly, I am not a expert of ML and AI, as a androgen receptor researcher, I think the research is interesting and may have potential use in predicting androgen receptor agonist and antagonist. Unfortunately, think the paper can't be published as it is.
The author introduced and compared several prediction method, but no clear conclusion, even no results showed in abstract.
A1.1: We thank the Reviewer for their comments that the research is interesting and may have potential use in predicting androgen receptor agonist and antagonist. The Abstract has now been rewritten in parts, so as to include results and conclusions. A Conclusions section has now been included, discussing the work and future directions: “
5. Conclusions
DNNs with 12 user-specified structure-, ligand- and statistically-based features were found to perform best at categorizing AR binders and nonbinders. They outperformed DNN with cddd features, as well as RF, and CNN methods, as well as regressors (expectedly, given the sharp category bins), as well as the same features in a multivariate logistical fashion, as well as simple docking. Implications are that explainability in ML features is important, as physicochemically- and biologically-relevant descriptors can perform best at the categorization for this particular AR dataset. In addition, different ML techniques may be best suited for different application tasks, with DNN performing better than RF given the overtraining seen in the RF models. CNN models may require more information, such as the protein-ligand binding pose or trajectories. The cddd featurizations may well perform better for property prediction and virtual screening. In the present work, the Chimp structure-based features performed better than other protein-derived features in this work and others published elsewhere. Improvement is still possible, given that the MCC can be higher for the evaluation compounds even if predictions obtained were good and improved on predictions for a golden standard of AR reference compounds. Better data, that is, less unbalanced and with better structural diversity may help improve future predictions, as could be combining the present features with other ML and non-ML techniques, such as boosting, or molecular dynamics simulations, respectively.“
R1.2: The results need some subtitles, at least a introduction sentence in each section is needed.
A1.2: Subtitles have now been added to each subsection in the Results section
R1.3: I strongly suggest the author add a conclusion paragraph at the end of the manuscript.
A1.3: A Conclusions section has now been included, discussing the work and future directions. See also A1.1
R1.4: The discussion section needs rewrite, I understand the author did some of discussions in results section. I suggest the author rewrite the discussion and move the discussion sentence from results, or put the results and discussion together.
A1.4: Parts of the Discussion have now been rewritten. In addition, a Conclusions section has now been included, discussing the work and future directions. See also A1.1
Reviewer 2 Report
Manuscript titled Ändrogen Receptor Binding Prediction with Random Forest, Deep Neural Networks, and Graph Convolutional Neural Networks" explored different modeling techniques for modeling AR modulating compounds. Overall, the work is well designed and executed but lacking a clear conclusion.
Author Response
Reviewer 2: Manuscript titled Ändrogen Receptor Binding Prediction with Random Forest, Deep Neural Networks, and Graph Convolutional Neural Networks" explored different modeling techniques for modeling AR modulating compounds. Overall, the work is well designed and executed but lacking a clear conclusion
A2.1: We thank the Reviewer for their comments that the work is well designed and executed. A Conclusions section has now been included, discussing the work and future directions :
“5. Conclusions
DNNs with 12 user-specified structure-, ligand- and statistically-based features were found to perform best at categorizing AR binders and nonbinders. They outperformed DNN with cddd features, as well as RF, and CNN methods, as well as regressors (expectedly, given the sharp category bins), as well as the same features in a multivariate logistical fashion, as well as simple docking. Implications are that explainability in ML features is important, as physicochemically- and biologically-relevant descriptors can perform best at the categorization for this particular AR dataset. In addition, different ML techniques may be best suited for different application tasks, with DNN performing better than RF given the overtraining seen in the RF models. CNN models may require more information, such as the protein-ligand binding pose or trajectories. The cddd featurizations may well perform better for property prediction and virtual screening. In the present work, the Chimp structure-based features performed better than other protein-derived features in this work and others published elsewhere. Improvement is still possible, given that the MCC can be higher for the evaluation compounds even if predictions obtained were good and improved on predictions for a golden standard of AR reference compounds. Better data, that is, less unbalanced and with better structural diversity may help improve future predictions, as could be combining the present features with other ML and non-ML techniques, such as boosting, or molecular dynamics simulations, respectively.“
Reviewer 3 Report
General
This manuscript should not be accepted it does not contain any new research in the area. In addition the results presented and discussed are too raw (premature) and are not processed enough with the ML tools used. I give some remarks below the authors can account for in a new revision or submission of the material in the future.
Specific remarks
- 1. Abstract: The abstract does not contain any specific new results and are instead written an introduction text. On the other hand, the last paragraph of introduction, where the aim of study normally is given, instead contains the results. So I suggest the authors re-arrange and modify the text accordingly.
- Introduction: An introduction should describe state of the art with a background what others has done in the field but also the own community, but the result should not be given. However, the last paragraph of introduction, from lines 47-57, contains conclusions from the results that I think fits better to Abstract, see point above. Here instead the Aim if the Study should be presented.
In the introduction it should be defined and clearly presented what is a binder/non-binder from a thermodynamic point of view. I do not think that was clearly described in all citations either, as example not in the current ref 16. What is the definition in digits in terms of distribution coefficients and rate constants for the equilibrium and the kinetics in the system, respectively? What is the definition in such quantitative terms of a strong binder and antagonist etcetera? The authors should show and raise awareness of the problem of the inherent heterogeneity of bindings, see as example description of the inherent multitude of heterogeneity here:
https://pubs.acs.org/doi/10.1021/acs.analchem.8b00504
https://pubs.acs.org/doi/10.1021/acs.analchem.0c02475
This was for the kinetics which in term leads to equilibrium but also for smaller molecules like here it does not exist homogenous interactions. It is only an assumption made to allow easier calculations. These references above should be added in the authors references to show how how complex this question is, and to raise awareness about it.
- 3. Results
- a) Tables 1-3 should be moved to a supplementary material /Appendix section.
- b) The digits in the x scale in Figure 5 are too crowded. One cannot distinguish these numbers.
- c) The Figures 6 a-c lacks units and data and the legends is so small so it is impossible see anything. These figures must be remade entirely.
- d) The 4.3 and 4.4 subsections under 4. Materials and Methods are very messy and needs to be streamlined accordingly.
Author Response
Reviewer 3: General. This manuscript should not be accepted it does not contain any new research in the area. In addition the results presented and discussed are too raw (premature) and are not processed enough with the ML tools used. I give some remarks below the authors can account for in a new revision or submission of the material in the future.
A3.1: We respectfully disagree with the Reviewer. The present work shows new DNN models with new, physicochemically- and biologically-relevant features that perform better than RF, CNN, other well-known DNN methods with other featurizations, simple docking, and other published studies on this, state-of-the-art data. Toxicology is a difficult task to predict, but inevitable and of critical importance to health and development of materials and useful substances. We have rewritten parts in all sections of the manuscript, as well as included better comparisons including new Figure 8 with Receiver-Operator Curves (ROC) and Area Under the Curve (AUC) values and comparison to other methods, as well as Discussion and Conclusions:
“5. Conclusions
DNNs with 12 user-specified structure-, ligand- and statistically-based features were found to perform best at categorizing AR binders and nonbinders. They outperformed DNN with cddd features, as well as RF, and CNN methods, as well as regressors (expectedly, given the sharp category bins), as well as the same features in a multivariate logistical fashion, as well as simple docking. Implications are that explainability in ML features is important, as physicochemically- and biologically-relevant descriptors can perform best at the categorization for this particular AR dataset. In addition, different ML techniques may be best suited for different application tasks, with DNN performing better than RF given the overtraining seen in the RF models. CNN models may require more information, such as the protein-ligand binding pose or trajectories. The cddd featurizations may well perform better for property prediction and virtual screening. In the present work, the Chimp structure-based features performed better than other protein-derived features in this work and others published elsewhere. Improvement is still possible, given that the MCC can be higher for the evaluation compounds even if predictions obtained were good and improved on predictions for a golden standard of AR reference compounds. Better data, that is, less unbalanced and with better structural diversity may help improve future predictions, as could be combining the present features with other ML and non-ML techniques, such as boosting, or molecular dynamics simulations, respectively.”
R3.2: Abstract: The abstract does not contain any specific new results and are instead written an introduction text. On the other hand, the last paragraph of introduction, where the aim of study normally is given, instead contains the results. So I suggest the authors re-arrange and modify the text accordingly.
A3.2: We thank the Reviewer for the comment. We have now rewritten the Abstract accordingly: “Substances that can modify the androgen receptor pathway in humans and animals are entering the environment and food chain with the proven ability to disrupt hormonal systems and leading to toxicity and adverse effects on reproduction, brain development, and prostate cancer, among others. State-of-the-art databases with experimental data of human, chimp, and rat effects by chemicals have been used to build machine learning classifiers and regressors and evaluate these on independent sets. Different featurizations, algorithms, and protein structures lead to different results, with deep neural networks (DNNs) on user-defined physicochemically-relevant features developed for this work outperforming graph convolutional, random forest, and large featurizations. The results show that these user-provided structure-, ligand- and statistically-based features and specific DNNs provided the best results as determined by AUC (0.87), MCC (0.47), and other metrics and by their interpretability and chemical meaning of the descriptors/features. In addition, the same features in the DNN method performed better than in a multivariate logistic model: validation MCC = 0.468 and training MCC = 0.868 for the present work compared to evaluation set MCC = 0.2036 and training set MCC = 0.5364 for the multivariate logistic regression on the full, unbalanced set. Techniques of this type may improve AR and toxicity description and prediction, improving assessment and design of compounds. Source code and data are available at https://github.com/AlfonsoTGarcia-Sosa/ML “
See also point 3.3 below
R3.3: Introduction: An introduction should describe state of the art with a background what others has done in the field but also the own community, but the result should not be given. However, the last paragraph of introduction, from lines 47-57, contains conclusions from the results that I think fits better to Abstract, see point above. Here instead the Aim if the Study should be presented.
A3.3: The last part of the Introduction has now been rewritten as: “The present work shows that different modeling techniques can have their advantages and disadvantages for modeling AR modulating compounds. Deep neural networks (DNNs) and graph convolutional neural networks (CNNs) have been used in other modeling studies, usually using featurization included in widely-available packages [17]. Here, an effort was made to build Random Forest (RF) and DNNs with a given set of features that are chemically important based on calculated protein-ligand binding to several targets, chemical fingerprint distances, and other results from statistical techniques. Aiming to improve the predicted categorization of chemical compounds as AR binders using physicochemically- and biologically-relevant features can help in flagging molecules that may have potential to disrupt AR pathways, and thus, may have the potential of toxic effects. In silico prediction of these effects is important given the reduction of animal testing, and the expense of testing, as well as a first, fast complement to testing” See also point 3.2 above
R3.4: In the introduction it should be defined and clearly presented what is a binder/non-binder from a thermodynamic point of view. I do not think that was clearly described in all citations either, as example not in the current ref 16. What is the definition in digits in terms of distribution coefficients and rate constants for the equilibrium and the kinetics in the system, respectively? What is the definition in such quantitative terms of a strong binder and antagonist etcetera? The authors should show and raise awareness of the problem of the inherent heterogeneity of bindings, see as example description of the inherent multitude of heterogeneity here:
https://pubs.acs.org/doi/10.1021/acs.analchem.8b00504
https://pubs.acs.org/doi/10.1021/acs.analchem.0c02475
This was for the kinetics which in term leads to equilibrium but also for smaller molecules like here it does not exist homogenous interactions. It is only an assumption made to allow easier calculations. These references above should be added in the authors references to show how how complex this question is, and to raise awareness about it.
A3.4: We thank the reviewers. These complex questions have now been discussed and references included in the Discussion section, page 16: “Evidently, there are considerations to be taken about how to classify kinetic data [21, 22], in many cases of biological interest, e.g., in antibody interactions, complex formation steady-state is not reached [21]; to distinguish between binding sites; and that analyzing interaction data from biosensor instruments is based on the simplified assumption that larger biomolecules interactions are homogeneous [22]. Also, that for the CoMPARA challenge [6], the organizers (EPA) used the thresholds determined in the CERAPP project and applied them to AR concentration-response values (AC50) from the literature, using the following scheme among several possible:
• Strong: Activity concentration < 0.09 microM
• Moderate: Activity concentration 0.09 – 0.18 microM
• Weak: Activity concentration 0.18 - 20 microM
• Very Weak: Activity concentration 20 - 800 microM
• Inactive: Activity concentration > 800 microM
The use of ML in the form of DNN with user-specified features on a balanced set provides better results as compared to the same features in a multivariate logistic fashion, as well as purely structure- or ligand-based approaches, as seen by better AUC, MCC, and other values.”
We would like to stress that we used the data as provided by the CoMPARA challenge, and the organizers themselves classified the binders and nonbinders based on the above criteria, which was the condition required for the challenge
R3.5: Results. a) Tables 1-3 should be moved to a supplementary material /Appendix section.
A3.5: We thank the Reviewer for this comment. However, these tables contain the data most relevant for the manuscript and moving them to the supplementary material /Appendix sections would remove all tables from the main text and make it harder to read and follow the material and text.
R3.6: b) The digits in the x scale in Figure 5 are too crowded. One cannot distinguish these numbers.
A3.6: Figure 5 has now been redone
R3.7: c) The Figures 6 a-c lacks units and data and the legends is so small so it is impossible see anything. These figures must be remade entirely.
A3.7: Figures 6 a-c have now been remade to be clearer and at larger size. Units have been included in the caption
R3.8:d) The 4.3 and 4.4 subsections under 4. Materials and Methods are very messy and needs to be streamlined accordingly.
A3.8: These subsections have now been revised and corrected
Reviewer 4 Report
The authors of the paper describe their proposed approach for Androgen Receptor Binding Prediction with Random Forest, Deep Neural Networks, and Graph Convolutional Neural Networks. The topic is interesting and with possible applicability. However, the paper needs several improvements:
1) the main contribution and originality should be explained in more detail
2) the motivation of the approach with CNNs needs further clarification
3) discussion of related work in neural networks should be expanded with more recent work
4) Minor grammar and syntax issues need correction
5) more simulation results and formal comparison of results are needed
6) the conclusions should be extended with more future work
7) More references to related papers in NNs should be included, like:
Optimal Feature Selection-Based Medical Image Classification Using Deep Learning Model in Internet of Medical Things. IEEE Access 8: 58006-58017 (2020)
Automatic Vehicle License Plate Recognition Using Optimal K-Means With Convolutional Neural Network for Intelligent Transportation Systems. IEEE Access 8: 92907-92917 (2020)
An Integrated Hybrid CNN-RNN Model for Visual Description and Generation of Captions. Circuits Syst. Signal Process. 39(2): 776-788 (2020)
A new approach for classifying coronavirus COVID-19 based on its manifestation on chest X-rays using texture features and neural networks. Inf. Sci. 545: 403-414 (2021)
Author Response
Reviewer 4: The authors of the paper describe their proposed approach for Androgen Receptor Binding Prediction with Random Forest, Deep Neural Networks, and Graph Convolutional Neural Networks. The topic is interesting and with possible applicability. However, the paper needs several improvements: 1) the main contribution and originality should be explained in more detail
A4.1: We thank the Reviewer for their comments that the topic is interesting and with possible applicability. The Abstract and parts of the Introduction have now been rewritten; especially the last paragraph contains these points. A Conclusions section has now been included, discussing the work and future directions. See also point A4.6 below
R4.2: 2) the motivation of the approach with CNNs needs further clarification
A4.2: CNNs were included only as a comparison tool, together with Random Forests, and regressors, to compare to the DNN classification models that gave the best results. DNN models with other features other than the user-specified features did not give as good results as the DNN with the physicochemically- and biologically-relevant features employed and designed for this study. Different ML and other techniques will have different results in different application cases. See also point A4.6 below
R4.3: 3) discussion of related work in neural networks should be expanded with more recent work
A4.3: More and newer references to ML and DNN have now been included in the Introduction, page 1
R4.4: 4) Minor grammar and syntax issues need correction
A4.4: These have now been addressed with revision tools
R4.5: 5) more simulation results and formal comparison of results are needed
A4.5: We have now included Receiver-Operator Curves with plots and Area Under the Curve values for the Human, Chimp, and Rat androgen receptor docking scores. They are shown as Figures 8a-c, as well as text in the manuscript. They show a good separation of true positive from false positives in the most important, i.e., initial parts of the curves. Their values are high, and the chimp protein again shows that it is the most suited with an AUC of 0.832, and an enrichment factor at 1% of 68.92. AUC for human was 0.797, and AUC for rat was 0.744.This has been included in the manuscript on page 11, lines 5-10, as well as on page 12: “Comparing our results to a structure-based approach by Trisciuzzi et al.[REF], they obtained the highest AUC of 0.76 for structures 2pnu and 2hvc, compared to 0.83 for the Chimp AUC in this work.”
R4.6: 6) the conclusions should be extended with more future work
A4.6: A Conclusions section has now been included, discussing the work and future directions: "5. Conclusions
DNNs with 12 user-specified structure-, ligand- and statistically-based features were found to perform best at categorizing AR binders and nonbinders. They outperformed DNN with cddd features, as well as RF, and CNN methods, as well as regressors (expectedly, given the sharp category bins), as well as the same features in a multivariate logistical fashion, as well as simple docking. Implications are that explainability in ML features is important, as physicochemically- and biologically-relevant descriptors can perform best at the categorization for this particular AR dataset. In addition, different ML techniques may be best suited for different application tasks, with DNN performing better than RF given the overtraining seen in the RF models. CNN models may require more information, such as the protein-ligand binding pose or trajectories. The cddd featurizations may well perform better for property prediction and virtual screening. In the present work, the Chimp structure-based features performed better than other protein-derived features in this work and others published elsewhere. Improvement is still possible, given that the MCC can be higher for the evaluation compounds even if predictions obtained were good and improved on predictions for a golden standard of AR reference compounds. Better data, that is, less unbalanced and with better structural diversity may help improve future predictions, as could be combining the present features with other ML and non-ML techniques, such as boosting, or molecular dynamics simulations, respectively."
See also point A4.1 above
R4.7: 7) More references to related papers in NNs should be included, like:
Optimal Feature Selection-Based Medical Image Classification Using Deep Learning Model in Internet of Medical Things. IEEE Access 8: 58006-58017 (2020)
Automatic Vehicle License Plate Recognition Using Optimal K-Means With Convolutional Neural Network for Intelligent Transportation Systems. IEEE Access 8: 92907-92917 (2020)
An Integrated Hybrid CNN-RNN Model for Visual Description and Generation of Captions. Circuits Syst. Signal Process. 39(2): 776-788 (2020)
A new approach for classifying coronavirus COVID-19 based on its manifestation on chest X-rays using texture features and neural networks. Inf. Sci. 545: 403-414 (2021)
A4.7: These references have now been included on page 1, line 58
Round 2
Reviewer 1 Report
The revised version has been greatly improved, I suggest the author increased the fond of labels in the following figures:
Figure 2, Figure 3, Figure 6 and Figure 7.
Author Response
We thank the Reviewer for their comment. We have now revised Figures 2, 3, and 5 with bigger fonts. Figures 6 and 7 are produced with other software that cannot be modified
Reviewer 3 Report
General
The revised manuscript has been considerably improved and the authors has also point by point carefully answered with sound motivations my points. Also other changes made, are fine. However some clarifications remains. I think the revised manuscript should be published after some minor revisions according to below.
Specific remarks
- Still the Aim of study, that should be in the last paragraph in introduction is not clear enough. I am referring to lines 67-72, “Aiming to improve ….testing”. The 2nd sentence starting at line 70 with “In silico..” is better. But as a whole this paragraph focus on explaining what is the benefit of this research. This is not what should be told here in the last paragraph even if the benefit is good to mention. It should be more particularly written what is the aim if this particular study. The conclusion section must harmonize with this "Aim of Study" text.
- Figure 5 is better. But I do think still the digits in the x scale could be made more readable, by making them a little larger. That could be done by using exponential numbers so that the many following zeroes could be lifted, giving enough space to use a somewhat larger font size. digits?
- The details in the legends in Figures 6 and 7 are still very small and difficult to read. Perhaps the signs could also here be enlarged some?
- The new paragraph in “3. Discussion” is really good for understanding what a such number means and verify the conditions selected.
- The new section “5. Conclusion” is also good. But perhaps god gave in mind some potential readers only read title + abstract + conclusion before deciding going deeper in the manuscript and by that reason it is good to avoid too many abbreviations explained in the manuscript. Also consider that the conclusion must in a way couple to, or harmonize to the aim of study mentioned above (see point 1).
Author Response
R3.1. The revised manuscript has been considerably improved and the authors has also point by point carefully answered with sound motivations my points. Also other changes made, are fine. However some clarifications remains. I think the revised manuscript should be published after some minor revisions according to below.
- Still the Aim of study, that should be in the last paragraph in introduction is not clear enough. I am referring to lines 67-72, “Aiming to improve ….testing”. The 2nd sentence starting at line 70 with “In silico..” is better. But as a whole this paragraph focus on explaining what is the benefit of this research. This is not what should be told here in the last paragraph even if the benefit is good to mention. It should be more particularly written what is the aim if this particular study. The conclusion section must harmonize with this "Aim of Study" text.
A3.1 We thank the Reviewer for their comment that the manuscript has been considerably improved. The last part of the Introduction (page 2, lines 18-21) and the Conclusions (page 18, lines 10-28) sections have now been rewritten. See also point A3.5 below
R3.2. Figure 5 is better. But I do think still the digits in the x scale could be made more readable, by making them a little larger. That could be done by using exponential numbers so that the many following zeroes could be lifted, giving enough space to use a somewhat larger font size. digits?
A3.2. Figure 5 has now been remade
R3.3. The details in the legends in Figures 6 and 7 are still very small and difficult to read. Perhaps the signs could also here be enlarged some?
A3.3. Figure 6 and 7 were made with different software that cannot be modified
R3.4. The new paragraph in “3. Discussion” is really good for understanding what a such number means and verify the conditions selected.
A3.4. We thank the Reviewer for their comment that the new paragraph in “3. Discussion” is really good for understanding what a such number means and verify the conditions selected
R3.5. The new section “5. Conclusion” is also good. But perhaps god gave in mind some potential readers only read title + abstract + conclusion before deciding going deeper in the manuscript and by that reason it is good to avoid too many abbreviations explained in the manuscript. Also consider that the conclusion must in a way couple to, or harmonize to the aim of study mentioned above (see point 1).
A3.5. We thank the Reviewer for their comment that the Conclusions section is good. The last part of the Introduction (page 2, lines 18-21) and the Conclusions (page 18, lines 10-28) sections have now been rewritten. See also point A3.1 above
Reviewer 4 Report
The authors have addressed all my concerns and the paper could be accepted.
Author Response
We thank the Reviewer for their comments that the concerns have been addressed and the paper can now be accepted